# STOCHASTIC INTERPOLANTS WITH DATA-DEPENDENT COUPLINGS

## ABSTRACT

Generative models inspired by dynamical transport of measure – such as flows and diffusions – construct a continuous-time map between two probability densities. Conventionally, one of these is the target density, only accessible through samples, while the other is taken as a simple base density that is data-agnostic. In this work, using the framework of stochastic interpolants, we formalize how to *couple* the base and the target densities, whereby samples from the base are computed conditionally given samples from the target in a way that is different from (but does preclude) incorporating information about class labels or continuous embeddings. This enables us to construct dynamical transport maps that serve as conditional generative models. We show that these transport maps can be learned by solving a simple square loss regression problem analogous to the standard independent setting. We demonstrate the usefulness of constructing dependent couplings in practice through experiments in super-resolution and in-painting.

## 1 INTRODUCTION

Generative models such as normalizing flows and diffusions sample from a target density $\rho_1$ by continuously transforming samples from a base density $\rho_0$ into the target. This transport is accomplished by means of an Ordinary Differential Equation (ODE) or Stochastic Differential Equation (SDE), which takes as initial condition a sample from $\rho_0$ and produces at time $t = 1$ an approximate sample from $\rho_1$. Typically, the base density is taken to be something simple, analytically tractable, and easy to sample, such as a standard Gaussian. In some formulations, such as score-based diffusion (Sohl-Dickstein et al., 2015; Song & Ermon, 2020; Ho et al., 2020b; Song et al., 2020; Singhal et al., 2023), a Gaussian base density is intrinsically tied to the process achieving the transport. In others, including flow matching (Lipman et al., 2022b; Chen & Lipman, 2023), rectified flow (Liu et al., 2022b; 2023b), and stochastic interpolants (Albergo & Vanden-Eijnden, 2022; Albergo et al.,

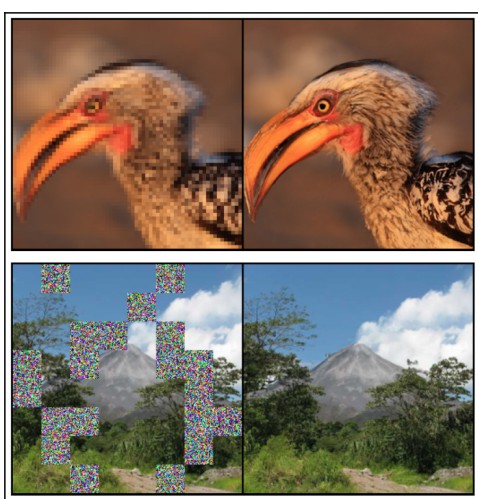

**Figure 1:** Example results from super-resolution and in-painting applications of stochastic interpolants with data-dependent couplings.

2023), a Gaussian base is not required, but is often chosen for convenience. In these cases, the choice of Gaussian base represents an absence of prior knowledge about the problem structure, and existing works have yet to fully explore the strength of base densities adapted to the target.

In this work, we introduce a general formulation of coupled and conditional base densities, building on the framework of stochastic interpolants. To do so, we introduce the notion of a base density produced via a *coupling*, whereby samples of the base are computed conditionally given samples from the target. We construct a continuous-time stochastic process that interpolates between the coupled

---

* Equal Contribution.

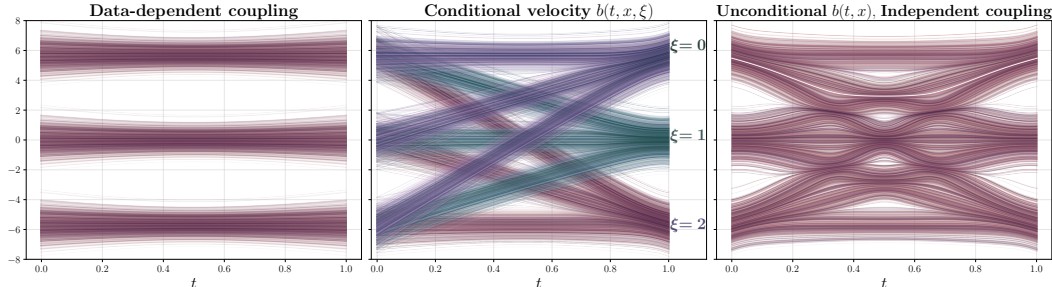

**Figure 2: Data-dependent couplings are different than conditioning.** Delineating between constructing couplings versus conditioning the velocity field, and their implications for the corresponding probability flow $X_t$. The transport problem is flowing from a Gaussian Mixture Model (GMM) with 3 modes to a GMM with 3 modes. *Left*: The probability flow $X_t$ arising from the data-dependent coupling $\rho(x_0, x_1) = \rho_1(x_1)\rho_0(x_0|x_1)$. All samples follow simple trajectories. No formation of auxiliary modes form in the intermediate density $\rho(t)$, in juxtaposition to the independent case. *Center*: When the velocity field is conditioned $b(t, x, \xi)$ on each class (mode), it factorizes, resulting in three separate probability flows $X_t^\xi$ with $\xi = 1, 2, 3$. *Right*: The probability flow $X_t$ when taking an unconditional velocity field $b(t, x)$ and an independent coupling $\rho(x_0, x_1) = \rho_0(x_0)\rho_1(x_1)$. Note the complexity of the underlying transport. This motivates us to consider finding such correlated base variables directly in the data.

base and target, and we characterize the resulting transport by identification of a continuity equation obeyed by the time-dependent density. We show that the velocity field defining this transport can be estimated by solution of an efficient, simulation-free square loss regression problem analogously to standard, data-agnostic interpolant and flow matching algorithms.

In our formulation, we also allow for dependence on an external, conditional source of information independent of $\rho_1$, which we call $\xi$. This extra source of conditioning is standard, and can be used in the velocity field $b(t, x, \xi)$ to accomplish class-conditional generation, or generation conditioned on a continuous embedding such as a textual representation or problem-specific geometric information. As illustrated in Fig. 2, it is however different from the data-dependent coupling that we propose. Below, we suggest some generic ways to construct coupled, conditional base and target densities, and we consider practical applications to image super-resolution and in-painting, where we find improved performance by incorporating both a data-dependent coupling and the conditional variable. Together, our **main contributions** can be summarized as:

1. We define a broader way of constructing base and target pairs in generative models based on dynamical transport that adapts the base to the target. In addition, we formalize the use of conditional information – both discrete and continuous – in concert with this new form of *data coupling* in the stochastic interpolant framework. As special cases of our general formulation, we obtain several recent variants of conditional generative models that have appeared in the literature.

2. We provide a characterization of the transport that results from conditional, data-dependent generation, and analyze theoretically how these factors influence the resulting time-dependent density

3. We provide an empirical study on the effect of coupling for stochastic interpolants. Given their recent promise as flexible generative models and the widespread recognition of the importance of conditioning and dependence in generative models, this represents a timely contribution.

4. We demonstrate the utility of data-dependent base densities and the use of conditional information in two canonical applications, image inpainting and super-resolution, which highlight the performance gains that can be obtained through the application of the tools developed here.

The rest of the paper is organized as follows. In Section 1.1, we describe some related work in conditional generative modeling. In Section 2, we introduce our theoretical framework. We characterize the transport that results from the use of data-dependent couplings, and discuss the difference between this approach and conditional generative modeling. In Section 3, we apply the

**Table 1: Couplings.** Standard formulations of flows and diffusions construct generative models built upon an independent coupling (Albergo & Vanden-Eijnden, 2022; Albergo et al., 2023; Lipman et al., 2022b; Liu et al., 2022b). Lee et al. (2023) learn $q_\phi(x_0|x_1)$ jointly with the velocity to define the coupling during training, but instead sample from $\rho_0 = \mathsf{N}(0, Id_d)$ for generation. Tong et al. (2023) and Pooladian et al. (2023) build couplings by sampling independently from the base and target and running mini-batch optimal transport algorithms (Cuturi, 2013). Here we focus on couplings enabled by our generic formalism, which bears similarities with Liu et al. (2023a), and can be individualized to each generative task.

| Coupling PDF $\rho(x_0, x_1)$ | Base PDF | Description |
| --- | --- | --- |
| $\rho_1(x_1)\rho_0(x_0)$ | $x_0 \sim \mathsf{N}(0, Id_d)$ | Independent |
| $\rho(x_0|x_1)\rho_1(x_1)$ | $x_0 \sim q_\phi(x_0|x_1)$ | Learned conditional |
| mb-OT$(x_1, x_0)$ | $x_0 \sim \mathsf{N}(0, Id_d)$ | Minibatch OT |
| $\rho_1(x_1)\rho_0(x_0|x_1)$ | $x_0 \sim \rho_0(x_0|x_1)$ | Dependent-coupling (**this work**) |

framework to numerical experiments on ImageNet, focusing on image inpainting and and image super-resolution. We conclude with some remarks and discussion in Section 4.

## 1.1 RELATED WORKS

**Couplings.** Several works have studied the question of how to build couplings, primarily from the specific viewpoint of optimal transportation theory. An initial perspective in this regard comes from Pooladian et al. (2023); Tong et al. (2023); Klein et al. (2023), who state an unbiased means for building entropically-regularized optimal couplings from minibatches of training samples. This perspective is appealing in that it may give probability flows that are straighter and hence more easily computed using simple ODE solvers. However, it relies on estimating an optimal coupling over minibatches of the entire dataset, which, for large datasets, may become uninformative as to the true coupling. In an orthogonal perspective, Lee et al. (2023) presented an algorithm to learn a coupling between the base and the target by building dependence on the target into the base. They argue that this can reduce curvature of the underlying transport. While this perspective allows them to empirically reduce the curvature of the flow lines, it introduces a potential bias in that they still sample from an independent base. Closely connected to our work is the approach proposed in Liu et al. (2023a): by considering generative modeling through the lens of Schrödinger bridges, they arrive after several approximations to a formulation that is operationally similar to, but less general than, ours. Our approach is simpler, and more flexible, as it differentiates between the bridging of the densities and the construction of the generative models. Table 1 summarizes these couplings along with the standard independent pairing. Other works also approach couplings from the perspective of Schrödinger bridges, with applications in, for example, biological problems (Somnath et al., 2023; Bunne et al., 2022).

**Generative Modeling and Dynamical Transport.** Generative models built upon dynamical transport of measure go back at least to (Tabak & Vanden-Eijnden, 2010; Tabak & Turner, 2013), and were further developed in (Rezende & Mohamed, 2015; Dinh et al., 2017; Huang et al., 2016; Durkan et al., 2019) using compositions of discrete maps, while modern models are typically formulated via a continuous-time transformation. In this context, a major advance was the introduction of score-based diffusion (Song et al., 2021b;a), which relates to denoising diffusion probabilistic models (Ho et al., 2020a), and allows one to generate samples by learning to reverse a stochastic differential equation that maps the data into samples from a Gaussian base density. Methods such as flow matching (Lipman et al., 2022a), rectified flow (Liu, 2022; Liu et al., 2022a), and stochastic interpolants (Albergo & Vanden-Eijnden, 2022; Albergo et al., 2023) expand on the idea of building stochastic processes that connect a base density to the target, but allow for bases that are more general than a Gaussian density. Typically, these constructions assume that the samples from the base and the target are uncorrelated.

**Conditional Diffusions and Flows for Images.** Saharia et al. (2022); Ho et al. (2022a) build diffusions for super-resolution, where low-resolution images are given as inputs to a score model,

which formally learns a conditional score (Ho & Salimans, 2022). In-painting can be seen as a form of conditioning where the conditioning set determines some coordinates in the target space. In-painting diffusions have been applied to video generation (Ho et al., 2022b) and protein backbone generation (Trippe et al., 2022). In the *replacement method* one directly inputs the clean values of the known coordinates at each step of integration (Ho et al., 2022b); Schneuing et al. (2022) replace with draws of the diffused state of the known coordinates. Trippe et al. (2022); Wu et al. (2023) discuss approximation error in this approach and correct with sequential Monte-Carlo. We revisit this problem framing from the velocity modeling perspective in Section 3.1. Recent work has started to apply flows for high dimensional conditional modeling (Dao et al., 2023; Hu et al., 2023). A Schrödinger bridge perspective on the conditional generation problem was presented in (Shi et al., 2022).

## 2 STOCHASTIC INTERPOLANTS WITH COUPLINGS

Suppose that we are given a dataset $\{x_1^i\}_{i=1}^n$. The aim of a generative model is to draw new samples assuming that the data set comes from a Probability Density Function (PDF) $\rho_1(x_1)$. Following the stochastic interpolant framework (Albergo & Vanden-Eijnden, 2022; Albergo et al., 2023), we introduce a time-dependent stochastic process that interpolates between samples from a simple base density $\rho_0(x_0)$ at time $t = 0$ and samples from the target $\rho_1(x_1)$ at time $t = 1$:

**Definition 1** (Stochastic interpolant with coupling). *The stochastic interpolant $x_t$ is the stochastic process defined as*[1]

$$x_t = \alpha(t)x_0 + \beta(t)x_1 + \gamma(t)z \qquad t \in [0, 1], \tag{1}$$

*where*

- *$\alpha(t)$, $\beta(t)$, and $\gamma^2(t)$ are differentiable functions of time such that $\alpha(0) = \beta(1) = 1$, $\alpha(1) = \beta(0) = \gamma(0) = \gamma(1) = 0$, and $\alpha^2(t) + \beta^2(t) + \gamma^2(t) > 0$ for all $t \in [0, 1]$.*

- *The pair $(x_0, x_1)$ are jointly drawn from a probability density $\rho(x_0, x_1)$ such that*

$$\int_{\mathbb{R}^d} \rho(x_0, x_1)dx_1 = \rho_0(x_0), \qquad \int_{\mathbb{R}^d} \rho(x_0, x_1)dx_0 = \rho_1(x_1). \tag{2}$$

- *$z \sim \mathsf{N}(0, Id)$, independent of $(x_0, x_1)$.*

A simple instance of (1) uses $\alpha(t) = 1 - t$, $\beta(t) = t$, and $\gamma(t) = \sqrt{2t(1-t)}$.

The stochastic interpolant framework uses information about the process $x_t$ to derive either an ODE or SDE whose solutions $X_t$ push the law of $x_0$ onto the law of $x_t$ for all times $t \in [0, 1]$.

As shown in Section 2.1, the drift coefficients in these ODEs/SDEs can be estimated by quadratic regression. They can then be used as generative models, owing to the property that the process $x_t$ specified in Definition 1 satisfies $x_{t=0} = x_0 \sim \rho_0(x_0)$ and $x_{t=1} = x_1 \sim \rho_1(x_1)$, and hence samples the desired target density. By drawing samples $x_0 \sim \rho_0(x_0)$ and using them as initial data $X_{t=0} = x_0$ in the ODEs/SDEs, we can then generate samples $X_{t=1} \sim \rho_1(x_1)$ via numerical integration.

In the original papers, this construction was made using the choice $\rho(x_0, x_1) = \rho_0(x_0)\rho_1(x_1)$, so that $x_0$ and $x_1$ were drawn independently from the base and the target.

*Our aim here is to build generative models that are more powerful and versatile by exploring and exploiting dependent couplings between $x_0$ and $x_1$ via suitable definition of $\rho(x_0, x_1)$.*

**Remark 1** (Incorporating conditioning). *Our formalism allows (but does not require) that each data point $x_1^i \in \mathbb{R}^d$ comes with a label $\xi_i \in D$, such as a discrete class or a continuous embedding like a text caption. In this setup, our results can be straightforwardly generalized by making all the quantities (PDF, velocities, etc.) conditional on $\xi$. This is discussed in Appendix A and used in our numerical examples.*

---

[1]More generally, we may set $x_t = I(t, x_0, x_1)$ in (1), where $I$ satisfies some regularity properties in addition to the boundary conditions $I(t = 0, x_0, x_1) = x_0$ and $I(t = 1, x_0, x_1) = x_1$ (Albergo & Vanden-Eijnden, 2022; Albergo et al., 2023). For simplicity, we will stick to the linear choice $I(t, x_0, x_1) = \alpha(t)x_0 + \beta(t)x_1$.

## 2.1 Transport equations and conditional generative models

In this section, we show that the probability distribution of the process $x_t$ defined in (1) has a time-dependent density $\rho(t, x)$ that interpolates between $\rho_0(x)$ and $\rho_1(x)$. We characterize this density as the solution of a transport equation, and we show that both the corresponding velocity field and the score $\nabla \log \rho(t, x)$ are minimizers of simple quadratic objective functions.

This result enables us to construct conditional generative models by approximating the velocity (and possibly the score) via minimization over a rich parametric class such as neural networks. We first define the functions

$$g_0(t, x) = \mathbb{E}(x_0 | x_t = x), \quad g_1(t, x) = \mathbb{E}(x_1 | x_t = x), \quad g_z(t, x) = \mathbb{E}(z | x_t = x) \tag{3}$$

where $\mathbb{E}(\cdot | x_t = x)$ denotes the expectation over $\rho(x_0, x_1)$ conditional on $x_t = x$. We then have,

**Theorem 2** (Transport equation with coupling). *The probability distribution of the stochastic interpolant $x_t$ defined in (1) has a density $\rho(t, x)$ that satisfies $\rho(t = 0, x) = \rho_0(x)$ and $\rho(t = 1, x) = \rho_1(x)$, and solves the transport equation*

$$\partial_t \rho(t, x) + \nabla \cdot (b(t, x)\rho(t, x)) = 0, \tag{4}$$

*where the velocity field can be written as*

$$b(t, x) = \dot{\alpha}(t)g_0(t, x) + \dot{\beta}(t)g_1(t, x) + \dot{\gamma}(t)g_z(t, x). \tag{5}$$

*Moreover, for every $t$ such that $\gamma(t) \neq 0$, the following identity for the score holds*

$$\nabla \log \rho(t, x) = -\gamma^{-1}(t)g_z(t, x). \tag{6}$$

*The functions $g_0$, $g_1$, and $g_z$ are the unique minimizers of the objectives*

$$L_0(\hat{g}_0) = \int_0^1 \mathbb{E}\left[|\hat{g}_0(t, x_t)|^2 - 2x_0 \cdot \hat{g}_0(t, x_t)\right] dt,$$

$$L_1(\hat{g}_1) = \int_0^1 \mathbb{E}\left[|\hat{g}_1(t, x_t)|^2 - 2x_1 \cdot \hat{g}_1(t, x_t)\right] dt, \tag{7}$$

$$L_z(\hat{g}_z) = \int_0^1 \mathbb{E}\left[|\hat{g}_z(t, x_t)|^2 - 2z \cdot \hat{g}_z(t, x_t)\right] dt,$$

*where $\mathbb{E}$ denotes an expectation over $(x_0, x_1) \sim \rho(x_0, x_1)$ and $z \sim \mathsf{N}(0, Id)$.*

A more general version of this result that includes a conditioning variable is proven in Appendix A. The objectives (7) can readily be estimated in practice from samples $(x_0, x_1) \sim \rho(x_0, x_1)$ and $z \sim \mathsf{N}(0, 1)$, which will enable us to learn approximations for use in a generative model. Moreover, because $\mathbb{E}(x_t | x_t = x) = x$ by definition, the functions $g_0$, $g_1$, and $g_z$ satisfy for every $t$ and $x$

$$\alpha(t)g_0(t, x) + \beta(t)g_1(t, x) + \gamma(t)g_z(t, x) = x. \tag{8}$$

This enables us to reduce computational expense: given two of the $g$'s, the third can always be calculated via (8). The transport equation (4) can be used to derive generative models, as we now show.

**Corollary 3** (Probability flow and diffusions with coupling). *The solutions to the probability flow equation*

$$\dot{X}_t = b(t, X_t, ) \tag{9}$$

*enjoy the property that*

$$X_{t=1} \sim \rho_1(x_1) \quad if \quad X_{t=0} \sim \rho_0(x_0) \tag{10}$$

$$X_{t=0} \sim \rho_0(x_0) \quad if \quad X_{t=1} \sim \rho_1(x_1) \tag{11}$$

*In addition, for any $\epsilon(t) \geq 0$, solutions to the forward SDE*

$$dX_t^F = b(t, X_t^F, )dt - \epsilon(t)\gamma^{-1}(t)g_z(t, X_t^F)dt + \sqrt{2\epsilon(t)}dW_t, \tag{12}$$

*enjoy the property that*

$$X_{t=1}^F \sim \rho_1(x_1) \quad if \quad X_{t=0}^F \sim \rho_0(x_0), \tag{13}$$

*and solutions to the backward SDE*

$$dX_t^R = b(t, X_t^R)dt + \epsilon(t)\gamma^{-1}(t)g_z(t, X_t^R)dt + \sqrt{2\epsilon(t)}dW_t, \tag{14}$$

*enjoy the property that*

$$X_{t=0}^R \sim \rho_0(x_0) \quad if \quad X_{t=1}^R \sim \rho_1(x_1). \tag{15}$$

A more general version of this result with conditioning is proven in Appendix A.

Let us now discuss a generic instantiation our formalism involving a specific choice of $\rho(x_0, x_1)$.

## 2.2 Data-dependent coupling

One natural way to allow for a data-dependent coupling between the base and the target is to set

$$\rho(x_0, x_1) = \rho_1(x_1)\rho_0(x_0|x_1) \quad \text{with} \quad \int_{\mathbb{R}^d} \rho_0(x_0|x_1)\rho_1(x_1)dx_1 = \rho_0(x_0). \tag{16}$$

There are many ways to construct the conditional $\rho_0(x_0|x_1)$. In the numerical experiments in Section 3.1 & Section 3.2, we consider base densities of the generic form

$$\rho_0(x_0|x_1) = \mathsf{N}(x_0; m(x_1), C(x_1)), \tag{17}$$

where the mean $m(x_1,) \in \mathbb{R}^d$ and covariance $C(x_1) \in \mathbb{R}^{d \times d}$ both depend on $x_1$. We stress that, even though the conditional $\rho_0(x_0|x_1)$ defined in (17) is a Gaussian density, $\rho(x_0, x_1) = \rho_1(x_1)\rho_0(x_0|x_1)$ and $\rho_0(x_0) = \rho_0(x_0|x_1)$ are non-Gaussian densities in general. Intuitively, the choice (17) constructs a coupling between samples $x_0$ and $x_1$ by applying a deterministic map to $x_1$ and corrupting the result with Gaussian noise whose variance also depends on $x_1$. A simple example is given by $m(x_1^i) = x_1$ and $C(x_1) = \sigma^2 Id$ for some $\sigma > 0$, which sets the base distribution to be a noisy version of the target.

## 2.3 Learning and Sampling

To learn in this setup, we can evaluate the objective functions (7) over a minibatch of $n_b < n$ data points $x_1^i$ by using an additional $n_b$ samples $z_i \sim \mathsf{N}(0, Id)$, $t_i \sim U([0, 1])$, and $x_0^i$ constructed as

$$x_0^i = m(x_1^i) + \sigma(x_1^i)\zeta_i, \tag{18}$$

with $\zeta_i \sim \mathsf{N}(0, Id_d)$ and $\sigma(x_1)\sigma^\top(x_1) = C(x_1)$. Setting $x_i = \alpha(t_i)x_0^i + \beta(t_i)x_0^i + \gamma(t_i)z_i$ then leads to the empirical approximation $\hat{L}_0$ of $L_0$ given by

$$\hat{L}_0(\hat{g}_0) = \frac{1}{n_b} \sum_{i=1}^{n_b} \left[ |\hat{g}_0(x_i)|^2 - 2x_0^i \cdot \hat{g}_0(x_i) \right], \tag{19}$$

with similar empirical variants for $L_1$ and $L_z$.

If we approximate the functions $g_0(t, x)$, $g_1(t, x)$, and $g_z(t, x)$ using a rich parametric class such as a set of neural networks, these empirical objectives can be minimized with respect to the parameters using an algorithm such as stochastic gradient descent. This leads to an approximation of the velocity $b(t, x)$ via (5) and an approximation of the score via (6).

Generating data requires sampling an $X_{t=0} \sim \rho_0(x_0)$ as an initial condition to be evolved via the probability flow ODE (9) or the forward SDE (12) to respectively produce a sample $X_{t=1} \sim \rho_1(x_1)$ or $X_{t=1}^F \sim \rho_1(x_1)$. Sampling an $x_0$ can be performed by picking data point $x_1$ either from the data set or from some online data acquisition procedure and using it in (18). The generated samples from either the probability flow ODE or forward SDE will be different from $x_1$, even with the choices $m(x_1) = x_1$ and $C(x_1) = \sigma^2 Id$. The probability flow ODE necessarily produces a single sample for each $x_0$, while the SDE produces a collection of samples whose spread can be controlled by the diffusion coefficient $\epsilon(t)$.

## 3 Numerical experiments

We now explore the proposed interpolant framework with dependent couplings on conditional image generation tasks; we find that the framework is straightforward to scale to high resolution images in pixel space.

## 3.1 In-painting

We consider an in-painting task, whereby $x_1 \in \mathbb{R}^{C \times W \times H}$ denotes an image with $C$ channels, width $W$, and height $H$. Given a pre-specified mask, the goal is to fill the pixels in the masked region with new values that are consistent with the entirety of the image. To fit this problem within our

data-dependent coupling framework, we set the conditioning variable $\xi \in \{0,1\}^{C \times W \times H}$ equal to the mask. For simplicity, we fix the values of the mask across channels, so that it corresponds to masking a specific region of the image in a channel-independent fashion. We define the base density by the relation $x_0 = \xi \circ x_1 + (1-\xi) \circ \zeta$, where $\circ$ denotes the Hadamard (elementwise) product and $\zeta \in \mathbb{R}^{C \times W \times H}, \zeta \sim \mathsf{N}(0, Id)$ denotes random white-noise values used to corrupt the pixels within the

**Table 2: FID for Inpainting Task.** FID comparison between under two paradigms: a baseline, where $\rho_0$ is a Gaussian with independent coupling to $\rho_1$, and our data-dependent coupling detailed in Section 3.1.

| Model | FID-50k |
|---|---|
| Uncoupled Interpolant (Baseline) | 1.35 |
| Dependent Coupling (**Ours**) | **1.13** |

masked region. Unlike the mask, these noise values can vary across channels. The spatial structure of the mask is drawn randomly by tiling the image into $64$ tiles; each tile is selected to enter the mask with probability $p = 0.3$. In our experiments, we set $\rho_1(x_1)$ to correspond to either Imagenet-256 or Imagenet-512. This corresponds to using $\rho(x_0, x_1|\xi) = \rho_1(x_1)\rho_0(x_0|x_1, \xi)$, which informs the model both of the masked regions and initializes the sample from the corrupted image. In the interpolant (1), we set $\alpha(t) = t$ and $\beta(t) = 1 - t$. In this setup, the optimal velocity field

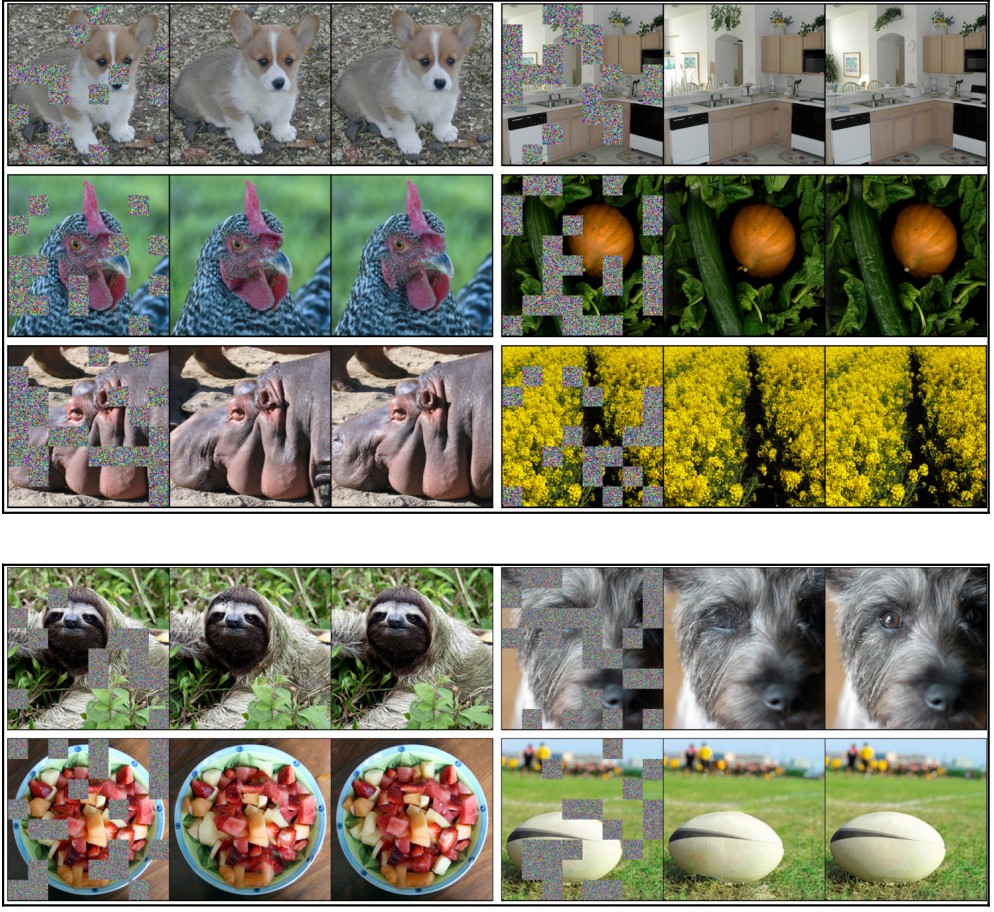

**Figure 3: Image inpainting: ImageNet-$256 \times 256$ and ImageNet-$512 \times 512$.** *Top panels*: Six examples of image in-filling at resolution $256 \times 256$, where the left columns display masked images, the center corresponds to in-filled model samples, and the right shows full reference images. The aims are not to recover the precise content of the reference image, but instead, to provide a conditionally valid in-filling. *Bottom panels*: Four examples at resolution $512 \times 512$.

$b(x, t, \xi) = b^*(x, t) \circ (1 - \xi)$ for a mask-independent velocity $b^*$. This follows because $\xi \circ x_t = \xi \circ x_1$ for every $t$, i.e., the unmasked pixels in $x_t$ are always those of $x_1$. To take this structural information

into account, we can build this property into our neural network model, and mask the output of the approximate velocity field to enforce that the unmasked pixels remain fixed.

**Results.** For implementation we learn a velocity model $b(x, t, y, \xi)$ where $y$ is the imagenet class label and $\xi$ is the missingness mask. In practice we append $\xi$ to the channels of the inputs $x$ which requires no modification to the basic Unet architecture from Ho et al. (2020b). Additional specific experimental details may be found in Appendix B. Samples are shown in Figure 3, as well as Section 1. FIDs are reported in Table 2. As discussed, the missing areas of the image are defined at time zero as independent normal random variables, depicted as colorful static in the images. In each image triple, the left panel is the base distribution sample $x_0$, the middle is the model sample of $X_{t=1}$ obtained by integrated the probability flow ODE (9), and the right panel is the ground truth. The generated textures, though different from the full sample, correspond to realistic samples from the conditional densities given the observed content.

### 3.2 SUPER-RESOLUTION ON IMAGENET

We now consider image super-resolution, in which we wish to take an image and convert it into an equivalent image at higher resolution. To this end, we let $x_1 \in \mathbb{R}^{C \times W \times H}$ correspond to a high-resolution image, as in Section 3.1. We denote by $\mathcal{D} : \mathbb{R}^{C \times W \times H} \to \mathbb{R}^{C \times W_{\text{low}} \times H_{\text{low}}}$ and $\mathcal{U} : \mathbb{R}^{C \times W_{\text{low}} \times H_{\text{low}}} \to \mathbb{R}^{C \times W \times H}$ image downsampling and upsampling operations, where $W_{\text{low}}$ and $H_{\text{low}}$ denote the width and

**Table 3: FID for Super-resolution, 64x64 to 256x256.**

| Model | FID-50K |
|---|---|
| Improved DDPM (Nichol & Dhariwal, 2021) | 12.26 |
| SR3 (Saharia et al., 2022) | 11.3 |
| ADM (Dhariwal & Nichol, 2021) | 7.49 |
| Cascaded Diffusion (Ho et al., 2022a) | 4.88 |
| I²SB (Liu et al., 2023a) | 2.70 |
| Dependent Coupling (**Ours**) | **2.13** |

height of a low-resolution image. To define the base density, we then set $x_0 = \mathcal{U}(\mathcal{D}(x_1)) + \sigma\zeta$ with $\zeta \in \mathbb{R}^{C \times W \times H}$, $\zeta \sim \mathsf{N}(0, Id)$, and $\sigma > 0$. Defining $x_0$ in this way frames the transport problem such that each starting pixel is proximal to its intended target. Notice in particular that, with $\sigma = 0$, each $x_0$ would correspond to a lower-dimensional sample embedded in a higher-dimensional space, and the corresponding distribution would be concentrated on a lower-dimensional manifold. Working with $\sigma > 0$ allows us to alleviate the associated singularities by adding a small amount of Gaussian noise to smooth the base density so it is well-defined over the entire higher-dimensional ambient space. In addition, we give the model access to the low-resolution image at all times; this problem setting then corresponds to using $\rho(x_0, x_1|\xi) = \rho_1(x_1)\rho_0(x_0|x_1, \xi)$ with $\xi = \mathcal{U}(\mathcal{D}(x_1))$. In the experiments, we set $\alpha(t) = t$ and $\beta(t) = 1 - t$, and we set $\rho_1$ to correspond to ImageNet-256 or ImageNet-512, following prior work (Saharia et al., 2022; Ho et al., 2022a).

**Results.** Similarly to the previous experiment, we append the upsampled low-resolution images $\xi$ to the channel dimension of the input $x$ of the velocity model, and likewise include the Imagenet class labels $y$. Samples are displayed in Fig. 4, as well as Section 1. Similar in layout to the previous experiment, the left panel of each triplet is the low-resolution image, the middle panel is the model sample $X_{t=1}$, and the right panel is the high-resolution image. The differences are easiest to see when zoomed-in. While the resolution increase from the model is very noticeable for 64 to 256, the differences even in ground truth images between 256 and 512 are more subtle. We also display FIDs for the 64x64 to 256x256 task, which has been studied in other works, in Table 3.

## 4 DISCUSSION, CHALLENGES, AND FUTURE WORK

In this work, we introduced a general framework for constructing data-dependent couplings between the base and target densities within the stochastic interpolant formalism. In addition, we highlighted how conditional information can be incorporated into the resulting transport by modifying both the base and the target. Dependent non-Gaussian base distributions have been explored implicitly in diffusion works that forego the stochastic differential equation formalism and stationary distribution requirements in place of general corruptions (Daras et al., 2022; Bansal et al., 2022), where generation can start from corrupted images; this motivates directly stating the modeling problem to start at such corruption, or more generally correlated base variables. We provide some suggestions for specific forms of data-dependent coupling, such as choosing for $\rho_0$ a Gaussian distribution with mean

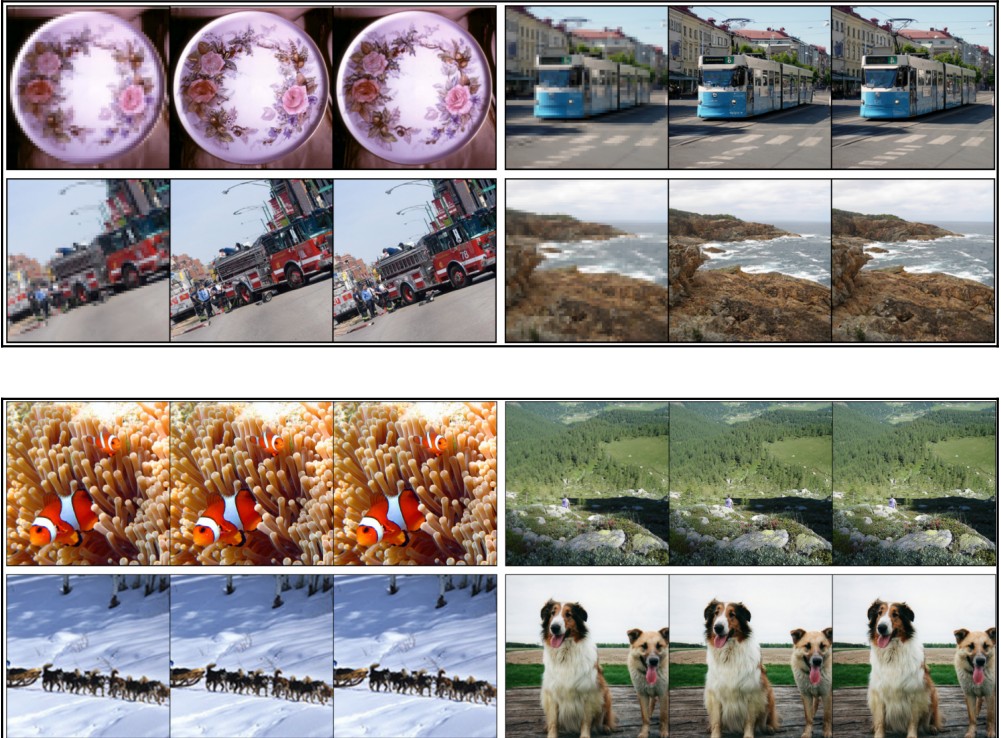

**Figure 4: Super-resolution:** *Top*: Super-resolved images from resolution $64 \times 64 \mapsto 256 \times 256$, where the left-most image is the lower resolution version, the middle is the model output, and the right is the ground truth. *Bottom*: The same procedure, but from $256 \times 256 \mapsto 512 \times 512$.

and covariance adapted to samples from the target, and showed how they can be used in practical problem settings such as image inpainting and super-resolution. There are many interesting generative modeling problems that stand to benefit from the incorporation of data-dependent structure. In the sciences, one potential application is in molecule generation, where we can imagine using data-dependent base distributions to fix a chemical backbone and vary functional groups. The dependency and conditioning structure needed to accomplish a task like this is similar to image inpainting. In machine learning, one potential application is in correcting autoencoding errors produced by an architecture such as a variational autoencoder (Kingma & Welling, 2013), where we could take the target density to be inputs to the autoencoder and the base density to be the output of the autoencoder.

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

## A  OMITTED PROOFS WITH CONDITIONING VARIABLES INCORPORATED

In this Appendix we give the proofs of Theorem 2 and Corollary 3 in a more general setup in which we incorporate conditioning variables in the definition of the stochastic interpolant.

To this end, suppose that each data point $x_1^i \in \mathbb{R}^d$ in the data set comes with a label $\xi_i \in D$, such as a discrete class or a continuous embedding like a text caption, and let us assume that this data set comes from a PDF decomposed as $\rho_1(x_1|\xi)\eta(\xi)$, where $\rho_1(x_1|\xi)$ is the density of the data $x_1$ conditioned on their label $\xi$, and $\eta(\xi)$ is the density of the label. In the following, we will somewhat abuse notation and use $\eta(\xi)$ even when $\xi$ is discrete (in which case, $\eta(\xi)$ is a sum of Dirac measures); we will however assume that $\rho_1(x_1|\xi)$ is a proper density. In this setup we can generalize Definition 1 as

**Definition 2** (Stochastic interpolant with coupling and conditioning). *The stochastic interpolant $x_t$ is the stochastic process defined as*

$$x_t = \alpha(t)x_0 + \beta(t)x_1 + \gamma(t)z \qquad t \in [0,1], \tag{20}$$

*where*

- *$\alpha(t)$, $\beta(t)$, and $\gamma^2(t)$ are differentiable functions of time such that $\alpha(0) = \beta(1) = 1$, $\alpha(1) = \beta(0) = \gamma(0) = \gamma(1) = 0$, and $\alpha^2(t) + \beta^2(t) + \gamma^2(t) > 0$ for all $t \in [0,1]$.*

- *The pair $(x_0, x_1)$ are jointly drawn from a conditional probability density $\rho(x_0, x_1|\xi)$ such that*

$$\int_{\mathbb{R}^d} \rho(x_0, x_1|\xi)dx_1 = \rho_0(x_0|\xi), \qquad \int_{\mathbb{R}^d} \rho(x_0, x_1|\xi)dx_0 = \rho_1(x_1|\xi). \tag{21}$$

- *$z \sim \mathsf{N}(0, Id)$, independent of $(x_0, x_1, \xi)$.*

Similarly, the functions (3) become

$$g_0(t, x, \xi) = \mathbb{E}(x_0|x_t = x), \quad g_1(t, x, \xi) = \mathbb{E}(x_1|x_t = x), \quad g_z(t, x, \xi) = \mathbb{E}(z|x_t = x) \tag{22}$$

where $\mathbb{E}(\cdot|x_t = x)$ denotes the expectation over $\rho(x_0, x_1|\xi)$ conditional on $x_t = x$, and Theorem 2 becomes:

**Theorem 4** (Transport equation with coupling and conditioning). *The probability distribution of the stochastic interpolant $x_t$ defined in (1) has a density $\rho(t, x|\xi)$ that satisfies $\rho(t = 0, x|\xi) = \rho_0(x|\xi)$ and $\rho(t = 1, x|\xi) = \rho_1(x|\xi)$, and solves the transport equation*

$$\partial_t \rho(t, x|\xi) + \nabla \cdot (b(t, x, \xi)\rho(t, x|\xi)) = 0, \tag{23}$$

*where the velocity field can be written as*

$$b(t, x, \xi) = \dot{\alpha}(t)g_0(t, x, \xi) + \dot{\beta}(t)g_1(t, x, \xi) + \dot{\gamma}(t)g_z(t, x, \xi). \tag{24}$$

*Moreover, for every $t$ such that $\gamma(t) \neq 0$, the following identity for the score holds*

$$\nabla \log \rho(t, x|\xi) = -\gamma^{-1}(t)g_z(t, x, \xi). \tag{25}$$

*The functions $g_0$, $g_1$, and $g_z$ are the unique minimizers of the objectives*

$$L_0(\hat{g}_0) = \int_0^1 \mathbb{E}\left[|\hat{g}_0(t, x_t, \xi)|^2 - 2x_0 \cdot \hat{g}_0(t, x_t, \xi)\right] dt,$$

$$L_1(\hat{g}_1) = \int_0^1 \mathbb{E}\left[|\hat{g}_1(t, x_t, \xi)|^2 - 2x_1 \cdot \hat{g}_1(t, x_t, \xi)\right] dt, \tag{26}$$

$$L_z(\hat{g}_z) = \int_0^1 \mathbb{E}\left[|\hat{g}_z(t, x_t, \xi)|^2 - 2z \cdot \hat{g}_z(t, x_t, \xi)\right] dt,$$

*where $\mathbb{E}$ denotes an expectation over $(x_0, x_1) \sim \rho(x_0, x_1|\xi)$, $\xi \sim \eta(\xi)$, and $z \sim \mathsf{N}(0, Id)$.*

Note that the objectives (26) can readily be estimated in practice from samples $(x_0, x_1) \sim \rho(x_0, x_1|\xi)$, $z \sim \mathsf{N}(0, 1)$, and $\xi \sim \eta(\xi)$, which will enable us to learn approximations for use in a generative model.

*Proof.* By definition of the stochastic interpolant given in (20), its characteristic function is given by

$$\mathbb{E}[e^{ik \cdot x_t}] = \int_{\mathbb{R}^d \times \mathbb{R}^d} e^{ik \cdot (\alpha(t)x_0 + \beta(t)x_1)} \rho(x_0, x_1 | \xi) dx_0 dx_1 e^{-\frac{1}{2} \gamma^2(t) |k|^2}, \tag{27}$$

where we used $z \perp (x_0, x_1)$ and $z \sim \mathsf{N}(0, Id_d)$. The smoothness in $k$ of (27) guarantees that the distribution of $x_t$ has a density $\rho(t, x | \xi) > 0$ globally. By definition of $x_t$, this density $\rho(t, x | \xi)$ satisfies, for any suitable test function $\phi : \mathbb{R}^d \to \mathbb{R}$,

$$\int_{\mathbb{R}^d} \phi(x) \rho(t, x | \xi) dx = \int_{\mathbb{R}^d \times \mathbb{R}^d \times \mathbb{R}^d} \phi(x_t) \rho(x_0, x_1 | \xi) (2\pi)^{-d/2} e^{-\frac{1}{2} |z|^2} dx_0 dx_1 dz. \tag{28}$$

Above, $x_t = \alpha(t)x_0 + \beta(t)x_1 + \gamma(t)z$. Taking the time derivative of both sides

$$
\begin{aligned}
&\int_{\mathbb{R}^d} \phi(x) \partial_t \rho(t, x | \xi) dx \\
&= \int_{\mathbb{R}^d \times \mathbb{R}^d \times \mathbb{R}^d} \left( \dot{\alpha}(t)x_0 + \dot{\beta}(t)x_1 + \dot{\gamma}(t)z \right) \cdot \nabla \phi(x_t) \rho(x_0, x_1 | \xi) (2\pi)^{-d/2} e^{-\frac{1}{2} |z|^2} dx_0 dx_1 dz \\
&= \int_{\mathbb{R}^d} \mathbb{E}\left[ \left( \dot{\alpha}(t)x_0 + \dot{\beta}(t)x_1 + \dot{\gamma}(t)z \right) \cdot \nabla \phi(x_t) \big| x_t = x \right] \rho(t, x | \xi) dx \\
&= \int_{\mathbb{R}^d} \mathbb{E}\left[ \dot{\alpha}(t)x_0 + \dot{\beta}(t)x_1 + \dot{\gamma}(t)z \big| x_t = x \right] \cdot \nabla \phi(x) \rho(t, x | \xi) dx
\end{aligned}
\tag{29}
$$

where we used the chain rule to get the first equality, the definition of the conditional expectation to get the second, and the fact that $\phi(x_t) = \phi(x)$ conditioned on $x_t = x$ to get the third. Since

$$\mathbb{E}\left[ \dot{\alpha}(t)x_0 + \dot{\beta}(t)x_1 + \dot{\gamma}(t)z \big| x_t = x \right] = \dot{\alpha}(t) g_0(t, x, \xi) + \dot{\beta}(t) g_1(t, x, \xi) + \dot{\gamma}(t) g_z(t, x, \xi) \tag{30}$$

by the definition of $g_0$, $g_1$, and $g_z$ in (22), we can use the definition of $b$ in (24) to write (29) as

$$\int_{\mathbb{R}^d} \phi(x) \partial_t \rho(t, x | \xi) dx = \int_{\mathbb{R}^d} b(t, x, \xi) \cdot \nabla \phi(x) \rho(t, x | \xi) dx. \tag{31}$$

This equation is (23) written in weak form.

To establish (25), note that if $\gamma(t) > 0$, we have

$$
\begin{aligned}
\mathbb{E}\left[ z e^{i\gamma(t)k \cdot z} \right] &= -\gamma^{-1}(t)(i\partial_k) \mathbb{E}\left[ e^{i\gamma(t)k \cdot z} \right], \\
&= -\gamma^{-1}(t)(i\partial_k) e^{-\frac{1}{2} \gamma^2(t) |k|^2}, \\
&= i\gamma(t) k e^{-\frac{1}{2} \gamma^2(t) |k|^2}.
\end{aligned}
\tag{32}
$$

As a result, using $z \perp (x_0, x_1)$, we have

$$\mathbb{E}\left[ z e^{ik \cdot x_t} \right] = i\gamma(t) k \mathbb{E}\left[ e^{ik \cdot x_t} \right]. \tag{33}$$

Using the properties of the conditional expectation, the left-hand side of this equation can be written

$$
\begin{aligned}
\mathbb{E}\left[ z e^{ik \cdot x_t} \right] &= \int_{\mathbb{R}^d} \mathbb{E}\left[ z e^{ik \cdot x_t} \big| x_t = x \right] \rho(t, x | \xi) dx, \\
&= \int_{\mathbb{R}^d} \mathbb{E}[z | x_t = x] e^{ik \cdot x} \rho(t, x, \xi) dx, \\
&= \int_{\mathbb{R}^d} g_z(t, x, \xi) e^{ik \cdot x} \rho(t, x, \xi) dx,
\end{aligned}
\tag{34}
$$

where we used the definition of $g_z$ in (22) to get the last equality. Since the right-hand side of (33) is the Fourier transform of $-\gamma(t) \nabla \rho(t, x | \xi)$, we deduce that

$$g_z(t, x, \xi) \rho(t, x | \xi) = -\gamma(t) \nabla \rho(t, x | \xi) = -\gamma(t) \nabla \log \rho(t, x | \xi) \rho(t, x | \xi). \tag{35}$$

Since $\rho(t, x | \xi) > 0$, this implies (25) when $\gamma(t) > 0$.

Finally, to derive (26), notice that we can write

$$
\begin{aligned}
L_0(\hat{g}_0) &= \int_0^1 \mathbb{E}\left[|\hat{g}_0(t, x_t, \xi)|^2 - 2x_0 \cdot \hat{g}_0(t, x_t, \xi)\right] dt, \\
&= \int_0^1 \int_{\mathbb{R}^d} \mathbb{E}\left[|\hat{g}_0(t, x_t, \xi)|^2 - 2x_0 \cdot \hat{g}_0(t, x_t, \xi)|x_t = x\right] \rho(t, x|\xi) dx dt \\
&= \int_0^1 \int_{\mathbb{R}^d} \left[|\hat{g}_0(t, x_t, \xi)|^2 - 2\mathbb{E}[x_0|x_t = x] \cdot \hat{g}_0(t, x, \xi)\right] \rho(t, x|\xi) dx dt \\
&= \int_0^1 \int_{\mathbb{R}^d} \left[|\hat{g}_0(t, x_t, \xi)|^2 - 2g_0(t, x, \xi) \cdot \hat{g}_0(t, x, \xi)\right] \rho(t, x|\xi) dx dt
\end{aligned}
\tag{36}
$$

where we used the definition of $g_0$ in (22). The unique minimizer of this objective function is $\hat{g}_0(x_t, \xi) = g_0(t, x, \xi)$, and we can proceed similarly to show that the unique minimizers of $L_1(\hat{g}_1)$ and $L_z(\hat{g}_z)$ are $\hat{g}_1(x_t, \xi) = g_1(t, x, \xi)$ and $\hat{g}_z(x_t, \xi) = g_z(t, x, \xi)$, respectively. □

Theorem 4 implies the following generalization of Corollary 3:

**Corollary 5** (Probability flow and diffusions with coupling and conditioning). *The solutions to the probability flow equation*

$$
\dot{X}_t = b(t, X_t, \xi)
\tag{37}
$$

*enjoy the property that*

$$
X_{t=1} \sim \rho_1(x_1|\xi) \quad \textit{if} \quad X_{t=0} \sim \rho_0(x_0|\xi)
\tag{38}
$$
$$
X_{t=0} \sim \rho_0(x_0|\xi) \quad \textit{if} \quad X_{t=1} \sim \rho_1(x_1|\xi)
\tag{39}
$$

*In addition, for any $\epsilon(t) \geq 0$, solutions to the forward SDE*

$$
dX_t^F = b(t, X_t^F, \xi)dt - \epsilon(t)\gamma^{-1}(t)g_z(t, X_t^F, \xi)dt + \sqrt{2\epsilon(t)}dW_t,
\tag{40}
$$

*enjoy the property that*

$$
X_{t=1}^F \sim \rho_1(x_1|\xi) \quad \textit{if} \quad X_{t=0}^F \sim \rho_0(x_0|\xi),
\tag{41}
$$

*and solutions to the backward SDE*

$$
dX_t^R = b(t, X_t^R, \xi)dt + \epsilon(t)\gamma^{-1}(t)g_z(t, X_t^R, \xi)dt + \sqrt{2\epsilon(t)}dW_t,
\tag{42}
$$

*enjoy the property that*

$$
X_{t=0}^R \sim \rho_0(x_0|\xi) \quad \textit{if} \quad X_{t=1}^R \sim \rho_1(x_1|\xi).
\tag{43}
$$

Note that if we additionally draw $\xi$ marginally from $\eta(\xi)$ when we generate the solution to these equations, we can also generate samples from the unconditional $\rho_0(x_0) = \int_D \rho_0(x_0|\xi)\eta(\xi)d\xi$ and $\rho_1(x_1) = \int_D \rho_1(x_1|\xi)\eta(\xi)d\xi$.

*Proof.* The probability flow ODE is the characteristic equation of the transport equation (23), which proves the statement about its solutions $X_t$. To establish the statement about the solution of the forward SDE (40), use expression (25) for $\nabla \log \rho(t, x, \xi)$ together with the identity $\Delta\rho(t, x, \xi) = \nabla \cdot (\nabla \log \rho(t, x, \xi) \rho(t, x, \xi))$ to write (23) as the forward Fokker-Planck equation

$$
\partial_t \rho(t, x|\xi) + \nabla \cdot \left((b(t, x, \xi) - \epsilon(t)\gamma^{-1}(t)g_z(t, x, \xi))\rho(t, x|\xi)\right) = \epsilon(t)\Delta\rho(t, x|\xi)
\tag{44}
$$

to be solved forward in time since $\epsilon(t) > 0$. To establish the statement about the solution of the reversed SDE (42), proceed similarly to write (23) as the backward Fokker-Planck equation

$$
\partial_t \rho(t, x|\xi) + \nabla \cdot \left((b(t, x, \xi) + \epsilon(t)\gamma^{-1}(t)g_z(t, x, \xi))\rho(t, x|\xi)\right) = -\epsilon(t)\Delta\rho(t, x|\xi)
\tag{45}
$$

to be solved backward in time since $\epsilon(t) > 0$. □

## B    FURTHER EXPERIMENTAL DETAILS

**Architecture**    For the velocity model we use the U-net from Ho et al. (2020b) as implemented in lucidrain's denoising-diffusion-pytorch repository; this variant of the architecture includes embeddings to condition on class labels. We use the following hyperparameters:

- Dim Mults: (1,1,2,3,4)
- Dim (channels): 256
- Resnet block groups: 8
- Leanred Sinusoidal Cond: True
- Learned Sinusoidal Dim: 32
- Attention Dim Head: 64
- Attention Heads: 4
- Random Fourier Features: False

**Image-shaped conditioning in the Unet**    For image-shaped conditioning, we follow Ho et al. (2022a) and append upsampled low-resolution images to the input $x_t$ at each time step to the velocity model. We also condition on the missingness masks for in-painting by appending them to $x_t$.

**Optimization**    . We use Adam optimizer (Kingma & Ba, 2014), starting at learning rate 2e-4 with the StepLR scheduler which scales the learning rate by $\gamma = .99$ every $N = 1000$ steps. We use no weight decay. We clip gradient norms at $10,000$ (this is the norm of the entire set of parameters taken as a vector, the default type of norm clipping in PyTorch library).

**Integration for sampling**    We use the Dopri solver from the torchdiffeq library (Chen, 2018).

**Miscellaneous**    We use Pytorch library along with Lightning Fabric to handle parallelism.

**Below we include additional experimental illustrations in the flavor of the figures in the main text.**

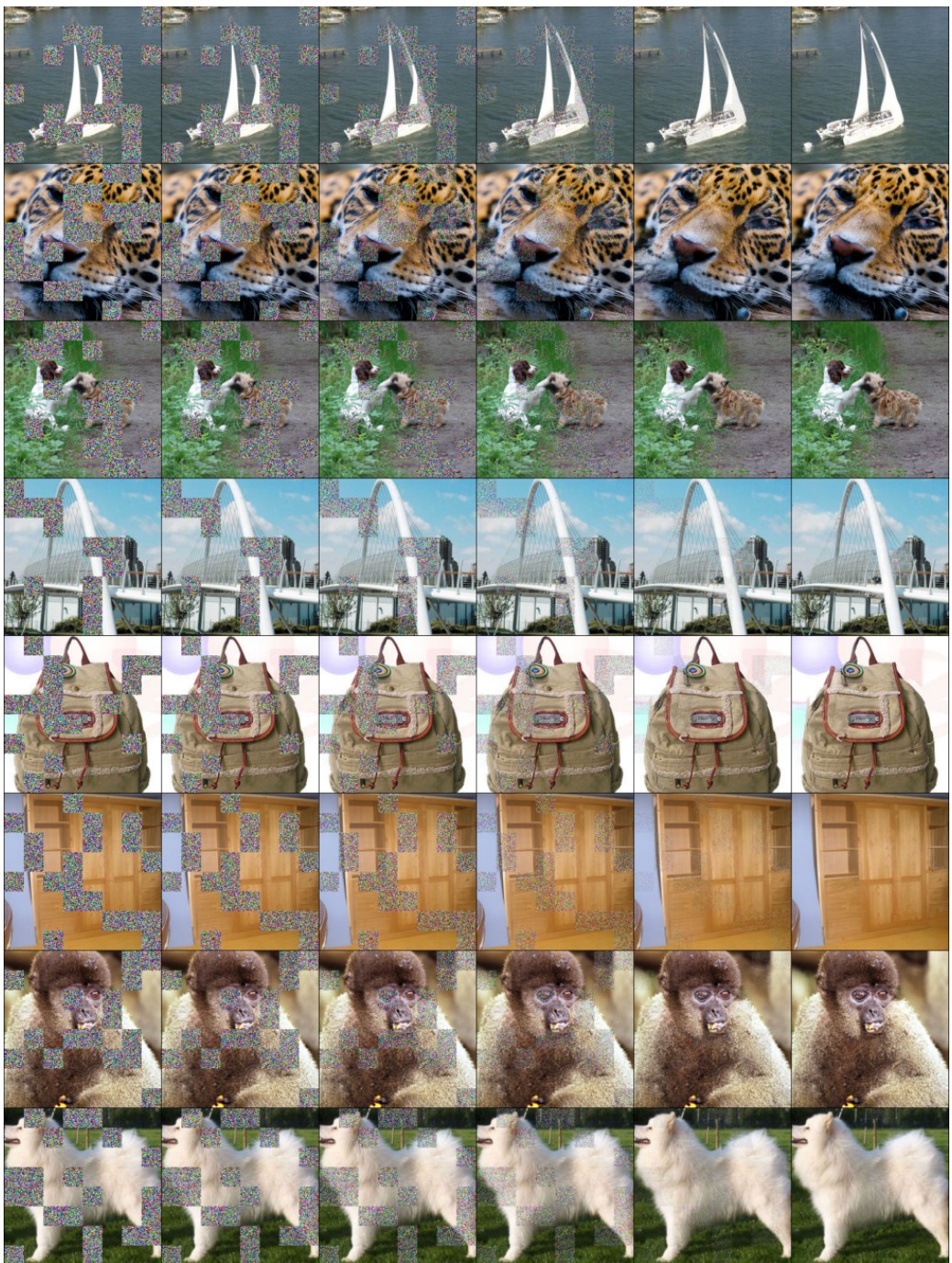

**Figure 5:** Additional examples of in-filling on the $256 \times 256$ resolution images, with temporal slices of the probability flow.

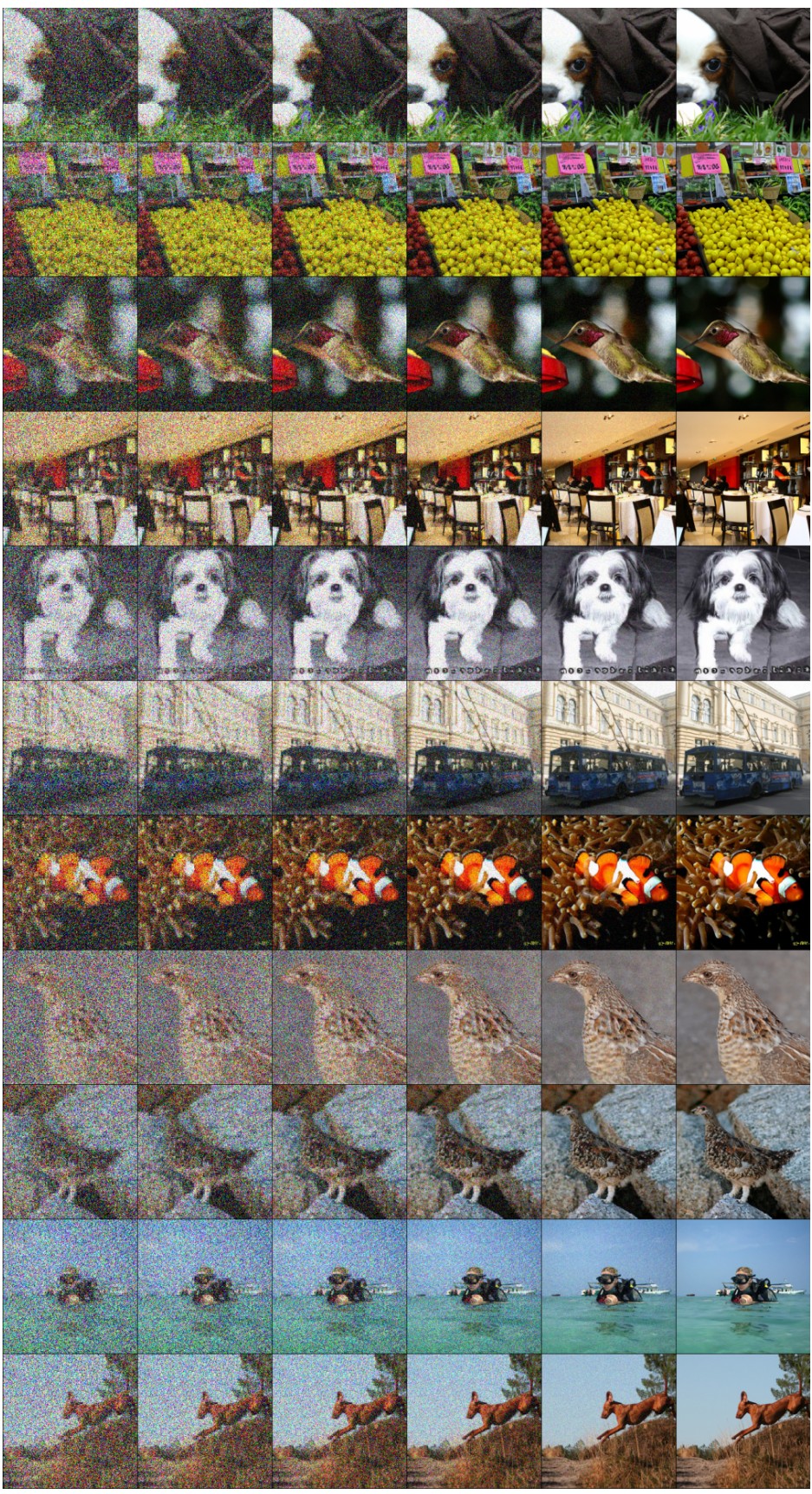

**Figure 6:** Additional examples of super-resolution from 64 to 256, with temporal slices of the probability flow.

