# OpenReview forum: "Stochastic interpolants with data-dependent couplings"
_ICLR.cc/2024/Conference — Submitted to ICLR 2024_

### Official Review · Reviewer_1uKk · 2023-10-23

**Soundness:** 3 good
**Presentation:** 3 good
**Contribution:** 2 fair
**Rating:** 6
**Confidence:** 4

**Summary:**

The paper introduces couplings between the prior and the target distribution in order to do (conditional) generative modelling. Here, plenty of former generative modelling frameworks are generalized. First the authors introduce a stochastic interpolant with coupling, then it is shown that the density satisfies the transport equation and a loss is derived. Furthermore, in the style of diffusion forward and reverse SDEs are introduced. Then this approach is applied to (class) conditional sampling for instance image superresolution and inpainting.

**Strengths:**

The idea is very neat and the theory seems well-executed. Qualitatively the experiments look really nice. Furthermore, the approach presents a nice unifying framework for many papers attempting to handcraft the couplings.

**Weaknesses:**

1) The biggest glaring weakness is the lack of quantitative experiments. While I am a huge fan of the idea and the developed theory, I think quantitative experimental evaluation is necessary for acceptance. An appropriate baseline could be the OT coupling flow from Tong et al.

2) In the proof of Theorem 1, I do not fully understand one step. In equation (23) for the second equality apparently the definition of conditional expectation is used. Please clarify this via some additional justification and the definition of conditional expectation. You kinda have to use that the expected value only depends on (the time derivative) of the stochastic coupling.

3) Is there any intuitive interpretation for the losses like in the diffusion case?

4) It would be nice to see failure cases of  joint learning of time coefficients and score. Since one simultaneously learns the time coefficients $\alpha$, $\beta$ .. and the $g$ I am expecting some not so nice local minima when one is not careful with initializing. Did you encounter any of those? Why didnt you use it in the inpainting/superres experiments and decided to fix $\alpha$ and $\beta$?

5) In the paper it is discussed, that one needs $\sigma>0$ in the superresolution experiment. If not one would try to establish a normalizing flow between a lower and a higher dimensional manifold. However when one thinks about the inpainting example filling all the boxes with random (Gaussian) noise could lead to an overestimation of the dimension of the target data, therefore prohibiting "a true transport equation" to hold as this defines a normalizing flow.

6) Is it possible to derive the losses from the forward/reverse SDEs so one does not necessarily enforce invertibility?

7) A small nitpick: I think the formulation "reverse" SDE is more appropriate since backward has a different meaning in the probability theory context.

**Questions:**

See weaknesses. Overall I appreciate the idea, but imo the following three things would greatly strengthen the paper: some quantitative evaluation, some discussion on the "invertibility" constraints and showing that learning of the schedules $\alpha,...$ also works in these image examples.

---

> ### Author Response · Authors · 2023-11-17
> **Really to reviewer 1uKk**
>
> We thank the reviewer for their input and address their main concerns one-by-one below. We also refer them to our general reply for some more information.
>
>
> ***Lack of quantitative experiments:*** We thank the reviewer for raising this point. We are currently running additional quantitative tests by comparing FID scores for our approach to existing baselines in the literature on super-resolution and in-painting. These results are shown in the table in our summary reply, where we achieve improved scores relative to earlier methods. We are continuing to train our models and expect the metrics to further improve for when we post our revision.
>
> ***Proof of Theorem 1:*** The penultimate step in Eq. (23) uses the the tower property of the conditional expectation. That is,
> $$\mathbb{E}[(\dot\alpha x_0+ \dot\beta x_1 + \dot\gamma z)\cdot \nabla \phi(x_t)] = \mathbb{E}[\mathbb{E}[(\dot\alpha x_0+ \dot\beta x_1 + \dot\gamma z)\cdot \nabla \phi(x_t)|x_t]].$$ The last step uses the fact that the conditioning of $\phi(x_t)$ on $x_t$ is trivial, i.e.
> $$ \mathbb{E}[\mathbb{E}[(\dot\alpha x_0+ \dot\beta x_1 + \dot\gamma z)\cdot \nabla \phi(x_t)|x_t]]
> = \mathbb{E}[\mathbb{E}[(\dot\alpha x_0+ \dot\beta x_1 + \dot\gamma z)|x_t]\cdot \nabla \phi(x_t)]$$
>
> ***Role of $\alpha(t)$, $\beta(t)$, and $\gamma(t)$:*** In the present paper, these functions of time are prescribed and are not optimized upon. The optimization is only performed over the parameters in the velocity field.
>
> ***Invertibility:*** The flow from base to target that we construct does not need to be invertible. In particular, it can map samples from a higher-dimensional manifold to samples on a lower-dimensonal one (though this comes at the expense of requiring a velocity that is singular at $t=1$). In contrast, traversing from a lower-dimensional manifold to a higher-dimensional one is more problematic, since it requires the velocity to be singular at $t=0$, and therefore there is ambiguity on how to start the flow. To avoid this latter scenario, we add a bit of noise to the sample from the base density that we use.
>
>
> ***Interpretation of the loss:*** One loss gives the conditional velocity appearing in the transport equation. The second loss gives the denoiser $\mathbb{E}[z|x_t=x]$, which allows us to estimate the score since $\nabla \log \rho(t,x) = - \gamma^{-1}(t) \mathbb{E}[z|x_t=x]$. That is, the loss for the denoiser $\mathbb{E}[z|x_t=x]$ can be viewed as a reweighted version of the loss for the score $\nabla \log \rho(t,x)$. This trick is exploited in diffusion models when learning a "noise model".
>
>
> **Reversed-time vs backward SDE:** We agree that the first denomination is better and we have adopted it.

---

> > ### Comment · Reviewer_1uKk · 2023-11-19
> > **Good update**
> >
> > Thanks for the response.
> >
> > Regarding the invertibility, this of course makes sense. As far as I understand your theory you basically need to assume Lebesgue densities for theorem 1 and corollary 2, right? In standard diffusion this is usually not needed, and there might be a way with your forward/reverse formulation to even derive a loss in this case.
> >
> > regarding the other points: the empirical evaluation seems now great, thanks!
> >
> > I am raising my score to 6. Please update the paper correspondingly, so I can check the updates.

---

> > > ### Author Response · Authors · 2023-11-20
> > > **Further clarifications on the theoretical question:**
> > >
> > > Thank you for your reply and for raising your score.
> > >
> > > To clarify the theoretical question: The current formulation of our Theorem 1 and Corollary 2 does require that the base and the target distributions have densities, in which case the flow is invertible. These results can be generalized to situations where we go from a higher dimensional manifold to a lower dimensional one, but in this case the drift must become singular at the end point (to focalize the process) whether we use an ODE or and SDE as generative process — our loss will  in principle learn this singular drift in this setup, rendering the flow noninvertible. This is similar to what happens with score based diffusion models since in the same setup the score also becomes singular at the end point (and plays the same focalizing role).  However, score-based diffusion models can also go from a lower dimensional manifold to a higher dimensional one, with the help of the diffusion term at initial time. In our approach, if we use a generative model based on an ODE, this cannot be achieved since the velocity would then have to be singular at initial time, and the flow cannot be started. We note that we could generalize our construction by using the SDE instead of the ODE as generative model  but this is not done here.  In practice, it seems to us that having the capability to focalize the process at final time (i.e. on the target distribution, since we always go forward in time from base to target) is what matters most.
> > >
> > > We will post a revised version of our paper before the end of the discussion period that will clarify these points and contain the result of our new numerical experiments.
> > >
> > > If there is anything else you want us to do that would potentially lead you to raise your score please let us know!

---

> > > > ### Author Response · Authors · 2023-11-23
> > > > **Thank you + revised draft posted**
> > > >
> > > > Dear Reviewer 1uKk,
> > > >
> > > > Thanks for helping us improve the quality of the work.
> > > >
> > > > As you requested, we have updated OpenReview with a new draft. This incorporates your suggestions and also includes some additional baselines for Improved DDPM and ADM.
> > > >
> > > > We hope that we have addressed your concerns and that you will consider raising your score (from marginal accept).
> > > >
> > > > Thank you!
> > > > Authors

---

### Official Review · Reviewer_RKsJ · 2023-10-31

**Soundness:** 4 excellent
**Presentation:** 3 good
**Contribution:** 1 poor
**Rating:** 6
**Confidence:** 4

**Summary:**

This paper extends the framework of stochastic interpolants to conditional generation. In particular, one conditions the interpolating density between $x_0$ and $x_1$ with a conditional $\xi$. The $\xi$ can be incorporated in a data dependent and independent way, which allows for applications in conditional generation as well as upsampling/infilling.

**Strengths:**

* The paper is theoretically sound, as the derivations follow directly from the continuity equation.
* Additionally, some experiments show the method's viability for common image generation tasks.

**Weaknesses:**

* I'm not sure the method is that original in practice. In particular, the paper notes that much of the construction can be connected with existing SDE and ODE formulations, all of which depend on the score function ([1] for the straight path ODE that is described in the paper, otherwise the standard OU process). In that case, the conditional methodology would follow from the score function argument as well, implying there would be little difference on the empirical side with existing methodologies. However, the proposed framework does generalize beyond this to other base distributions (for example), so I would expect (or rather, like to see) more empirical emphasis to be placed on this setting.
* In a similar vein, for the inpainting experiments, there is a big issue in that existing score based methods (e.g. ScoreSDE) can inpaint (up to some approximation error + some necessary hacks) without having to retrain, while the current results come about through retraining.
* The experiments don't give me that much confidence. In particular, the results are entirely qualitative (meaning they can be easily cherrypicked). For the upsampling experiments, I want to see some numerical comparisons against the standard cascaded diffusion models setup (eg generate 64x64 and upscale to 256x256 to compare FIDs).

[1] https://arxiv.org/abs/2303.00848

**Questions:**

Nothing beyond addressing the weaknesses.

---

> ### Author Response · Authors · 2023-11-17
> **Reply to reviewer RKsJ**
>
> We thank the reviewer for their input and we address their main concerns one-by-one below. We also refer them to our general reply for further information.
>
> ***Data-dependent coupling vs. class-conditioning:*** We stress that there is a distinct difference between putting conditional information in the velocity/score field and learning a map between coupled densities. For a visual explanation, see the figure posted [**here**](https://drive.google.com/file/d/1J04tNwIAkgaHXFpGSgse96o1WXGs2igc/view?usp=drive_link). The point of our paper is that data-dependent couplings can offer advantages that are orthogonal to (but can be combined with) conditional information placed in the velocity field. To emphasize this point, we have removed the conditional variable $\xi$ from the main text, and have clarified how it can be used with our coupling framework in the appendix.
>
> ***Originality:*** Our method with data-dependent coupling is general, as the base density $\rho(x_0|x_1)$ can be designed in many different ways. This offers a degree of design flexibility that is not available to existing approches.
>
> ***Need to retrain in the in-painting experiments:*** We would kindly like to request some clarification on the precise meaning of "retraining". In the response below, we have assumed that this refers to "classifier" or "guidance"-based methods for diffusion models, which can leverage a pre-trained model for conditioning. If this is incorrect, please let us know and we will provide an additional reply.
>
> To this end, we would first like to emphasize that we focus here on data-dependent couplings, as emphasized in our summary reply above. Nevertheless, to improve performance in super-resolution and in-painting, we *also* leverage conditional information. We would like to point out that conditioning can be used to improve performance in our method, but is not strictly necessary. For example, for in-painting, our coupling pairs a masked image with its unmasked counterpart. By contrast, a guidance-free diffusion model would likely perform much worse because it would have no information about the mask.
>
> We would also like to stress that our conditioning is over *high-dimensional images*, rather than the more standard scalar class label leveraged in guidance-based approaches. This means that the corresponding guidance model used in score-based methods is a **second** high-dimensional generative model, and its training is **comparably expensive** to the problem we solve here. In this sense, we would like to emphasize that our method does not require "retraining", as guidance-based methods would require an equivalent training effort.
>
> As stated above, please let us know if this is not what was meant by "retraining", and we will provide further explanation.
>
> ***Confidence in the experiments:*** We have added the requested FID benchmarks and see that they outperform the existing approaches.

---

> > ### Comment · Reviewer_RKsJ · 2023-11-18
> > **Found New Experiments Compelling,**
> >
> > Thanks for including the new experiments, which show that the method is able to achieve much better superresolution results than previous methods. Perhaps a comparison with ADM (Dhariwal and Nichol 2021) is applicable, since they also do 64x64->256x256 superresolution.
> >
> > For the inpainting, can there be a comparison with the method from Song et al 2021 (ScoreSDE)? I am not sure if there is a public checkpoint for diffusion models on ImageNet256x256, but perhaps we can compare on a smaller ImageNet scale (like 64x64) using a public checkpoint for an ImageNet UNet.
> >
> > In light of the experimental results for super-resolution, I am tentatively raising my score to a 6.

---

> ### Author Response · Authors · 2023-11-20
> **Comment on the  suggested experiments**
>
> Thank you for the feedback!
>
> For ***super-resolution***, [Ho et al 2021] (Cascading Diffusion Models, CDM) actually compares against the [Nichol and Dhariwal, 2021] (Improved DDPM) and [Dhariwal and Nichol, 2021] (ADM/guided-diffusion) models mentioned. For 64x64->256x256, [Ho et a 2021] reports FID scores of 12.26 for improved DDPM; 7.49 for ADM; and 4.88 for CDM.  ****The method we propose gives an FID score of 2.13****.  While the numbers here are reported from the tables in [Ho et al 2021], we will be glad to additionally run versions of them in our codebase over the coming weeks and report both the original and reproduced numbers.
>
> Regarding ***inpainting benchmarks***: after reading through the works you suggested, we found that [Song et al 2021] do not train any Imagenet models and that [Song, Durkan, et al 2021] only train on Imagenet 32x32. We have not trained any models at the lower resolutions of 32x32 and 64x64, so we will need time to do this.
>
> We will work on these additional benchmarks. In the meantime, we hope that our internal benchmark, in which we compare the results of our approach with data-coupling to those obtained using the interpolant framework with conditioning only and no data-coupling (similar to what score-based diffusion models would use), is informative: using only conditioning  performs well visually, but with worse FID score (1.35) than what we get with data-conditioned (**1.13**): see the table in the main reply above).
>
> [Ho et a 2021] Cascaded Diffusion Models for High Fidelity Image Generation
>
> [Nichol and Dhariwal, 2021] Improved Denoising Diffusion Probabilistic Models
>
> [Dhariwal and Nichol, 2021] Diffusion Models Beat GANs on Image Synthesis
>
> [Song et al 2021] Score-based Generative Modeling through Stochastic Differential Equations
>
> [Song, Durkan, et al 2021]  Maximum Likelihood Training of Score-Based Diffusion Models

---

> > ### Author Response · Authors · 2023-11-23
> > **Thank you + revised draft posted**
> >
> > Dear Reviewer RKsJ,
> >
> > Thanks for helping us improve the quality of the work. As mentioned, we also added the FIDs for Improved DDPM and ADM. We have updated OpenReview with an new draft incorporating your suggestions.
> >
> > We hope that we have addressed your concerns and that you will consider raising your score (from marginal accept).
> >
> > Thank you!
> > Authors

---

### Official Review · Reviewer_v6gN · 2023-11-01

**Soundness:** 3 good
**Presentation:** 3 good
**Contribution:** 2 fair
**Rating:** 5
**Confidence:** 4

**Summary:**

The paper formalizes conditional and data-dependent generative modeling within the stochastic interpolates framework. The authors derive the relevant transport equation for the deterministic scenario (ODE) and the forward and backward SDE for the stochastic scenario. They demonstrate that these equations can be acquired by minimizing straightforward regression losses. Lastly, data-dependent coupling is introduced, providing a recipe for constructing base densities that depend on the target distribution.

**Strengths:**

- The paper formalizes two important notions in generative modeling, conditional and data dependent coupling, in the stochastic interpolates framework.
- The authors show how to construct both conditional and data-dependent coupling.

**Weaknesses:**

1. **Limited contribution** - the work does not introduce a new concept and is a formulation of existing concepts into an existing framework.

   1. The derivation of the transport equations in section 2.1, which takes a great portion of the paper, was already done in section 4 of [3] for the unconditional case, where the addition of the conditioning repeats the same derivation with marginalization over the condition. Furthermore, conditioning for super-resolution has been shown in [5] as well as beening widely used in diffusion models (e.g., [4]), and since they can be thought of as particular cases of stochastic interpolants, the addition of conditioning is straightforward.
   2. Data dependent coupling was already introduced in the context of Flow-Matching [1,2], which is an essentially equivalent framework to stochastic interpolants.

While the work provides a coherent, complete formulation of conditional and data-dependent generative modeling in the stochastic interplant framework, I believe the paper needs to be reframed and further emphasize the analogies to existing works and highlight the benefits of formulating these concepts in the stochastic interpolants framework as opposed to for example Flow-matching which already provides the same degrees of flexibility in the design of generative models, or another example, the inpainting application considered in section 3.1 which is equivalent to the setting used in [4] only with a different noise scheduling.


2. **Empirical evaluation** - the empirical evaluation is solely qualitative, which makes it impossible to assess whether there is a benefit in using conditional and data dependent couplings in the stochastic interpolant framework.


[1] Pooladian et. al., Multisample Flow Matching: Straightening Flows with Minibatch Couplings (2023)

[2] Tong et. al., Improving and generalizing flow-based generative models with minibatch optimal transport (2023)

[3] Albergo et. al., Stochastic Interpolants: A Unifying Framework for Flows and Diffusions (2023)

[4] Saharia et. al., Palette: Image-to-Image Diffusion Models (2022)

[5]. Lipman et. al., Flow Matching for Generative Modeling (2022)

**Questions:**

1. Can the authors emphasize the analogies to existing works and highlight the benefits of formulating conditioning and data-dependent coupling concepts in the stochastic interpolants framework?
2. Does this formulation add flexibility compared to [1,2] or [6] which uses both conditioning and data dependent coupling (sec 5.3). (I'm aware [6] is not to be considered as previous work, but I want to understand the superiority of this work over applications already present in other frameworks).

[6] Song et. al., [Equivariant Flow Matching with Hybrid Probability Transport for 3D Molecule Generation](https://openreview.net/pdf?id=hHUZ5V9XFu)

---

> ### Author Response · Authors · 2023-11-17
> **Reply to reviewer v6gN**
>
> We thank the reviewer for their input and we address their main concerns one-by-one below. We also refer them to our general reply for further information.
>
> ***Derivation:***
> - While class-conditioning is indeed easily incorporated into the interpolant framework, coupling the base density to the data is a more significant change compared to the work in [3]. The resulting derivation of the transport equation follows similar steps, but the way to construct the coupled density $\rho_0(x_0,x_1) = \rho_0(x_0|x_1) \rho_1(x_1)$ via proper definition of $\rho_0(x_0|x_1)$ is non-trivial, as emphasizd in the next point.
>
> ***Data-dependent coupling vs mini-batch OT:***
> - The data-dependent coupling introduced here is general and can be used with any suitable $\rho(x_0,x_1) = \rho_0(x_0|x_1) \rho_1(x_1)$, which can be tailored to the application domain. In the experiments, we focus on examples in which the base $\rho(x_0 | x_1)$ is Gaussian conditional on the data $x_1$, but we stress that both the coupled density $\rho(x_0, x_1)$ and the marginal density $\rho_0(x_0) = \int \rho(x_0,x_1) dx_1$ are *not* Gaussian even in this case.
> - While related work by [1] and [2] showed how to construct minibatch couplings that try to approximate the *optimal* coupling, their perspective does not give tools to perform the more general types of data-dependent modeling we consider here. Moreover, the couplings we introduce are exact, and do not require (but also do not preclude) the use of additional algorithms such as Sinkhorn.
>
> We believe that these features make the approach we propose more flexible and performant than existing methods. In the revision, we will rework our presentation to clarify these points. We will also notify the reviewer as soon as the revised version becomes available.
>
> ***Empirical evaluation:***
> - As mentioned in our general reply, we have added FID benchmarks to demonstrate the utility of our data-dependent coupling.
>
> **Questions:**
> > *Can the authors emphasize the analogies to existing works and highlight the benefits of formulating conditioning and data-dependent coupling concepts in the stochastic interpolants framework?*
> - To re-emphasize our previous points, the main novelty of our work is to propose a generic way to perform data-dependent coupling. To do so, we exploit the flexibility in the base density afforded by the stochastic interpolant framework. This is different from class-conditioning (though the two approaches can be used in conjunction) and it offers a degree of design flexibility that is not available to existing methods.
>
> > *Does this formulation add flexibility compared to [1,2] or [6] which uses both conditioning and data dependent coupling (sec 5.3). (I'm aware [6] is not to be considered as previous work, but I want to understand the superiority of this work over applications already present in other frameworks).*
> - For our reply about references [1] and [2], please see the second bullet under ***Data-dependent coupling vs mini-batch OT***.
> - In the recent reference [6], to the best of our knowledge, the coupling proposed is another OT coupling akin to that of [1] and [2]. While the iterative algorithm presented there is interesting, and while we will surely cite this work, it remains a somewhat orthogonal to the general method we propose here.
>
> [1] [Pooladian et. al., Multisample Flow Matching: Straightening Flows with Minibatch Couplings (2023)](https://arxiv.org/abs/2304.14772)
>
> [2][Tong et. al., Improving and generalizing flow-based generative models with minibatch optimal transport (2023)](https://arxiv.org/abs/2302.00482)
>
> [3] [Albergo et. al., Stochastic Interpolants: A Unifying Framework for Flows and Diffusions (2023)](https://arxiv.org/abs/2303.08797)
>
> [4] [Saharia et. al., Palette: Image-to-Image Diffusion Models (2022)](https://arxiv.org/abs/2111.05826)
>
> [5] [Lipman et. al., Flow Matching for Generative Modeling (2022)](https://arxiv.org/abs/2210.02747)
>
> [6] [Song et. al., Equivariant Flow Matching with Hybrid Probability Transport for 3D Molecule Generation](https://openreview.net/pdf?id=hHUZ5V9XFu)

---

> ### Comment · Reviewer_v6gN · 2023-11-21
> **Follow up question**
>
> I thank the authors for their response.
>
> **Data-dependent coupling vs mini-batch OT:** I better understand the subtleties now. [1,2] focus on constructing the coupling satisfying both marginal constraints, i.e., define $\rho(x_0,x_1)$ such that $\rho_0(x) = \int \rho(x_0,x_1) dx_1$, $\rho_1(x) = \int \rho(x_0,x_1) dx_0$ given $\rho_0$ and $\rho_1$. In this work, the coupling is assumed to be a given  $\rho(x_0,x_1)$ and the paper shows how to construct $\rho_0$ implicitly by defining $\rho(x_0|x_1)$ given the coupling.
>
> The above perspective drives me to the realms of image to image translation applications (as also shown in the paper) but I believe there is not enough discussion on the relations to that. A very closely related paper with a great overlap in the basic ideas is [7]. Although framed under the Schrödinger Bridge formulation, to my understanding, the concepts are very similar except for this work smoothing with a gaussian the base $\rho(x_0|x_1)$. Furthermore, [7] has already shown the benefits of changing the base distribution over conditioning (like you showed in the additional experiments now).
> **Can the authors comment on the relation to this work?**
>
> In that case, I stand by my statement that I believe the contribution of this paper is limited.
>
> I do appreciate the formulation in another framework and I think the stochastic interpolants framework is simple and elegant, but this paper either needs to be reframed as a "formulation paper" and not as one introducing new concepts and adding a thorougher discussion on existing works or show strong experimental evaluation and ablations for the benefits of the subtle differences proposed in this work.
>
>
>
>
> [7] [Liu et. al., $I^2SB$: Image-to-Image Schrödinger Bridge (ICML 2023)](https://arxiv.org/abs/2302.05872)

---

> > ### Author Response · Authors · 2023-11-21
> > **Comparison with I2SB**
> >
> > We thank the reviewer for their reply, and for the link to reference [7], which we were unaware of. As rightly pointed out, the approach in [7] is related to ours and there is overlap in the application domains that we target (such as image super-resolution and image in-painting). In the revision, we will add a reference to [7] and a discussion of its content in light of our approach. As explained next, despite the similarities with [7], our viewpoint on the problem is distinct and leads to a simpler, more general, and more flexible formulation.
> >
> > ***In theory:*** The authors in [7] build a stochastic bridge between $\rho_0$ and $\rho_1$ by approximating the Schrödinger bridge between these two densities: this bridge is also meant to be used as generative process in the procedure. Constructing this approximation involves many steps, but, as most clearly seen in Equation (11) of reference [7], it eventually leads to using an interpolant
> > $$x_t = \alpha(t)x_0 + \beta(t)x_1 + \gamma(t)z$$
> > where the parameters $\alpha, \beta$, and $\gamma$ are linked by the relations:
> > $$\alpha(t) = \frac{\bar{\sigma}_t^2}{\bar{\sigma}_t^2 + \sigma_t^2},\qquad \beta(t) = \frac{\sigma_t^2}{\bar{\sigma}_t^2 + \sigma_t^2},\qquad \gamma(t) = \frac{\sigma_t\bar{\sigma}_t}{\sqrt{\bar{\sigma}_t^2 + \sigma_t^2}}.$$
> > Our results show that this link is unnecessary. *Any* interpolant with correlated pairs $(x_0,x_1)$, independent $z\sim\mathsf{N}(0,Id)$, and an arbitrary (up to suitable boundary conditions and differentiability requirements) set of $\alpha(t)$, $\beta(t)$, and $\gamma(t)$ forms a valid bridge between $\rho_0$ and $\rho_1$.  This more general formulation facilitates the design of flexible and performant generative models. Specifically:
> >
> > 1. Our approach does not require the bridge process to be a diffusion (let alone be an approximation of the Schrödinger bridge process, which is hard to solve). Instead it may be specified *explicitly*, from the get-go, via an interpolant, and therefore represents a more flexible and efficient class of exact bridges between $\rho_0$ and $\rho_1$.
> > 2. Our approach decouples the definition of the bridge from the construction of the generative process, thereby  enabling more flexible sampling procedures, including those based on diffusions (with a tunable diffusion coefficient that does not need to be related to the coefficients $\alpha$, $\beta$, and $\gamma$) or probability flow equations.
> >
> > ***In Experiment:*** our results show that the extra level of generality in our approach is useful in practice, and enables us to obtain FID scores that are lower than those reported in [7] (which unlike our approach, uses as intialization a fully trained state-of-the-art checkpoint, ADM). In particular (for a more complete table with comparisons to other works, see the new general reply above):
> >
> >
> > | Task            | Paper Citation        | FID Score           |
> > |-----------------|-----------------------|---------------------|
> > | Super Resolution| I2SB [7]               | 2.70              |
> > | Super Resolution| DD-Interpolant (**Ours**)  | **2.13**           |
> >
> >
> > We feel that the added simplicity of the derivations in our approach would, on its own, justify publication of our paper as a piece of work complementary to [7]. However, our work additionally shows that this simplicity leads to more design flexibility, and demonstrates its utility via the increased performance we achieve in the numerical examples compared to [7].
> >
> > Please do let us know if the reply above clarifies the question you had in regards to the conection betwen our work and [7], and please do ask if you have additional questions about the theory or the numerics.
> >
> > [7] Liu et. al., : Image-to-Image Schrödinger Bridge (ICML 2023)

---

> > > ### Author Response · Authors · 2023-11-23
> > > **Thank you + new draft posted**
> > >
> > > Dear Reviewer v6gN,
> > >
> > > Thanks for helping us improve the quality of the work.
> > >
> > > We have updated OpenReview with the new draft, which incorporates FIDs, discussion of I2SB, as well as your additional suggestions regarding clarity.
> > >
> > > We hope that we have addressed your concerns and that you will consider raising your score (from marginal reject).
> > >
> > > Thank you!
> > > Authors

---

### Author Response · Authors · 2023-11-17
**General reply 1/2**

We thank the reviewers for their valuable feedback on our paper, which has proven useful in clarifying our main contributions, their relations to prior work, and how to best demonstrate the advantages and flexibility of what we propose. Below, we summarize the feedback you have given us and what steps we have taken to improve the paper. Please let us know if we have misinterpreted your comments in any way. We look forward to hearing from you as you help us assemble an improved exposition for our work. Thank you again.

### **Summary of the reviewers' remarks:**
The reviewers observed that the theoretical contributions of our paper are sound (v6gN) and well-executed (1uKk), and that our paper formalizes conditional and data-dependent generative modeling within the stochastic interpolants framework (RKsJ). Reviewer RKsJ also noted the method’s viability for common image generation tasks.

However, reviewers v6gN and RKsJ questioned the novelty of our approach, and wanted us to better stress the differences between what it offers and what can be done with diffusion. They also wanted us to explain how our approach differs from the conditional generation tasks of [3, 5]. We provide these essential clarifications below.

In addition, all reviewers requested more rigorous quantitative verification of the method, which we will happily supply in the revision. As a teaser, we report here FID scores for in-painting and super-resolution which show that our method outperforms related and prior work at lower computational cost:

| Task            | Paper Citation        | FID Score           |
|-----------------|-----------------------|---------------------|
| Super Resolution| SR3 [3]               | 11.3                |
| Super Resolution| Cascaded diffusion [4]| 4.88 (with data augmentation)         |
| Super Resolution| DD-Interpolant (**Ours**)  | **2.13**           |
| | | |
| Inpainting      | baseline (from Gaussian) | 1.35             |
| Inpainting      | DD-Interpolant (**Ours**)| **1.13**             |
| | | |


### **Summarizing our results:**
Given a target density $\rho_1(x_1)$ available through data, we construct a *coupled* density $\rho(x_0,x_1) = \rho_0(x_0|x_1) \rho_1(x_1) \not = \rho_0(x_0) \rho_1(x_1)$ by *conditioning on the data* to define the base density $\rho(x_0 | x_1)$. We use this coupling to derive generative models based on probability flow ODEs or SDEs using the stochastic interpolant framework.


*We stress that there is a distinct difference between putting conditional information in the velocity/score field and learning a map between coupled densities. For a visual explanation, see the figure* **[here](https://drive.google.com/file/d/1J04tNwIAkgaHXFpGSgse96o1WXGs2igc/view?usp=drive_link)**. *The point of our paper is that data-dependent couplings can offer advantages that are orthogonal to (but can be combined with) conditional information placed in the velocity field. To emphasize this point, we have removed the conditional variable $\xi$ from the main text, and have clarified how it can be used with our coupling framework in the appendix.*

We also stress that using data-dependent coupling requires the ability to use an *arbitrary* base distribution, a feature made possible by the interpolant framework, but which is absent in methods that requires the base to be a Gausian density.

We break down this difference with respect to the existing flow-matching, in-painting, and super-resolution work in part 2 of this reply.

---

> ### Author Response · Authors · 2023-11-17
> **General reply 2/2**
>
> ### **Clarifying our contributions compared to prior and concurrent works:**
>
> **Clarifying our theoretical contributions:**
> - The data-dependent coupling introduced here is general and can be used with any suitable $\rho(x_0,x_1) = \rho_0(x_0|x_1) \rho_1(x_1)$, which can be tailored to the application domain. In the experiments, we focus on examples in which the base $\rho(x_0 | x_1)$ is Gaussian conditional on the data $x_1$, but we stress that both the coupled density $\rho(x_0, x_1)$ and the marginal density $\rho_0(x_0) = \int \rho(x_0,x_1) dx_1$ are *not* Gaussian even in this case.
> - While related work by [1] and [2] showed how to construct minibatch couplings that try to approximate the *optimal* coupling, their perspective does not give tools to perform the more general types of data-dependent modeling we consider here. Moreover, the couplings we introduce are exact, and do not require (but also do not preclude) the use of additional algorithms such as Sinkhorn.
> - This work extends previous results on stochastic interpolants to expound on the utility of using an arbitary base distribution.
>
>
> **Clarifying our experimental contributions:**
> Here we delineate the differences between the experiments we have presented and prior work on in-painting and super-resolution. Our primary point is that the use of a data-dependent base density has significant numerical advantages over uncoupled approaches, even if they make use of conditional information.
>
>
> - **Super-resolution:** Reviewer v6gN pointed to [3] and [4] as examples of work that have already implemented super-resolution. We emphasize that these are suitable *baselines* for our method. In both papers, a flow or a diffusion is used to push a Gaussian sample to a super-resolution image, where the vector field is conditioned on a low-resolution image. This is standard conditional generation. Instead, we propose to use a density in which each low resolution image is *coupled to its high-resolution counterpart*, rather than just to use a velocity field that is conditional on the low-resolution image. The advantage of this second approach is apparent from the FID table above.
> - **In-painting:** Here, we again use a dependent density as our base distribution, where we couple each masked image to its unmasked counterpart. This has the advantage that the velocity field can be taken as zero outside the mask, which allows for controlled generation. We show the benefit of this in compared to an uncoupled, standard normal base density in our numerics, where we see an improved FID.
>
>
> ### **Adjustments made to the exposition**
>
> Thanks to the feedback from the reviewers, we have adjusted the exposition in the text to highlight the subtleties in the differences of our method compared to prior works (as detailed above). Moreover, we have adjusted the numerical experiments, some preliminary results of which are discussed below and summarized in the table above. We will update the paper with the final numbers at the end of the rebuttal period.
>
>
>
> ### **New Numerical Results: FIDs**
>
> As per the requests of the reviewers, we have included Frechet Inception Distance (FID) benchmarks of our model vs others.
>
> **In-painting**: We compare FIDs for in-painting when generating from the coupled density $\rho(x_0 | x_1)$ against the baseline of generating from a standard normal on ImageNet $256\times256$. With only 200,000 steps of training, we achieve and FID of 1.13, compared to the baseline of 1.38.
>
> **Superresolution**: With only 300k steps of training, we achieve an FID for super-resolution of ImageNet 64x64 to ImageNet 256x256 of 2.13, which surpasses other known methods such as those mentioned by the reviewers.
>
>
> ### **References**
>
> [1] Pooladian et. al., Multisample Flow Matching: Straightening Flows with Minibatch Couplings (2023)
>
> [2]Tong et. al., Improving and generalizing flow-based generative models with minibatch optimal transport (2023)
>
> [3] Saharia et al: Image Super-Resolution via Iterative Refinement https://arxiv.org/abs/2104.07636  (2021)
>
> [4] Ho et al: Cascaded Diffusion Models for High Fidelity Image Generation, https://arxiv.org/abs/2106.15282 (2021)
>
> [5] Lipman et al: Flow matching for generative modeling, https://arxiv.org/abs/2210.02747 (2023)

---

> > ### Author Response · Authors · 2023-11-21
> > **Updated table with FID scores**
> >
> > | Task            | Paper Citation        | FID Score           |
> > |-----------------|-----------------------|---------------------|
> > | Super Resolution| Improved DDPM [6]               | 12.26                |
> > | Super Resolution| SR3 [3]               | 11.3                |
> > | Super Resolution| ADM [7]               | 7.49                |
> > | Super Resolution| Cascaded diffusion [4]| 4.88 (with data augmentation)         |
> > | Super Resolution| I2SB [8]               | 2.70              |
> > | Super Resolution| DD-Interpolant (**Ours**)  | **2.13**           |
> > |-----------------|-----------------------|---------------------|
> > | Inpainting      | baseline (from Gaussian) | 1.35             |
> > | Inpainting      | DD-Interpolant (**Ours**)| **1.13**             |
> >
> > [1] Pooladian et. al., Multisample Flow Matching: Straightening Flows with Minibatch Couplings (2023)
> >
> > [2]Tong et. al., Improving and generalizing flow-based generative models with minibatch optimal transport (2023)
> >
> > [3] Saharia et al: Image Super-Resolution via Iterative Refinement https://arxiv.org/abs/2104.07636  (2021)
> >
> > [4] Ho et al: Cascaded Diffusion Models for High Fidelity Image Generation, https://arxiv.org/abs/2106.15282 (2021)
> >
> > [5] Lipman et al: Flow matching for generative modeling, https://arxiv.org/abs/2210.02747 (2023)
> >
> > [6] Nichol and Dhariwal: [Improved Denoising Diffusion Probabilistic Model](https://arxiv.org/abs/2102.09672) (2021)
> >
> > [7] Dhariwal and Nichol: [Diffusion Models Beat GANs on Image Synthesis](https://arxiv.org/abs/2105.05233) (2021)
> >
> > [8] Liu et al: I2SB: [Image-to-Image Schrodinger Bridge](https://arxiv.org/abs/2302.05872) (2023)

---

### Author Response · Authors · 2023-11-23
**Thank you + updated draft (general reply)**

Dear Reviewers and Area Chair,

Thank you all for your valuable feedback.

We have uploaded a new draft, which incorporates the reviewers' suggestions. In summary, the changes focus around (1) updating tables and discussion to include more connections and numerical comparisons with related work (SR3, CDMs, Improved DDPM, ADM, I2SB, and others) and (2) clarifying contributions to focus on the couplings and how it differs from previous work.

We hope that these changes have addressed the primary concerns, and believe the discussion period has benefited the work immensely.

The reviewer have updated their scores from (3, 5, 5) to (6, 6, 5). We have provided discussion + baselines to address the new feedback and would appreciate consideration of whether the additional concerns have been addressed.

Thank you!
Authors

---

### Meta-Review · Area_Chair_em5n · 2023-12-11

**Metareview:**

This paper generalizes Stochastic Interplants (SI) to the case of arbitrary couplings of base and target distributions $(x_0,x_1)$. Applications to image inpainting and super-resolution are presented. This paper was discussed among reviewers during the discussion period and the 2 (out of total of 3) reviewers that participated in the discussion agreed that the conceptual contribution over previous work is slightly below the bar for acceptance at this point. In particular: A) the general coupling of base-target is already introduced [Pooladian et. al. 2023 (eq. 13,14)] and [Liu et al, 2023 (Table 1)], and B) [Liu et al. 2023] already utilized the problem-dependent coupling such as $x_0$ is a low res version of $x_1$. Given A+B the main contributions of the current submission are the improved FID in the super-resolution task, but this seems a bit too narrow to support acceptance at this time.

**Justification For Why Not Higher Score:**

Detailed in the meta review.

**Justification For Why Not Lower Score:**

N/A

---

### Decision · Program_Chairs · 2024-01-16

Reject